# GNSS Multipath Detection Using Continuous Time-Series C/N_0_

**DOI:** 10.3390/s20144059

**Published:** 2020-07-21

**Authors:** Nobuaki Kubo, Kaito Kobayashi, Rei Furukawa

**Affiliations:** Department of Maritime Systems Engineering, Tokyo University of Marine Science and Technology, Tokyo 135-8533, Japan; m195007@edu.kaiyodai.ac.jp (K.K.); rei-furukawa@kke.co.jp (R.F.)

**Keywords:** GNSS, multipath, NLOS, survey, DGNSS, RTK

## Abstract

The reduction of multipath errors is a significant challenge in the Global Navigation Satellite System (GNSS), especially when receiving non-line-of-sight (NLOS) signals. However, selecting line-of-sight (LOS) satellites correctly is still a difficult task in dense urban areas, even with the latest GNSS receivers. This study demonstrates a new method of utilization of C/N_0_ of the GNSS to detect NLOS signals. The elevation-dependent threshold of the C/N_0_ setting may be effective in mitigating multipath errors. However, the C/N_0_ fluctuation affected by NLOS signals is quite large. If the C/N_0_ is over the threshold, the satellite is used for positioning even if it is still affected by the NLOS signal, which causes the positioning error to jump easily. To overcome this issue, we focused on the value of continuous time-series C/N_0_ for a certain period. If the C/N_0_ of the satellite was less than the determined threshold, the satellite was not used for positioning for a certain period, even if the C/N_0_ recovered over the threshold. Three static tests were conducted at challenging locations near high-rise buildings in Tokyo. The results proved that our method could substantially mitigate multipath errors in differential GNSS by appropriately removing the NLOS signals. Therefore, the performance of real-time kinematic GNSS was significantly improved.

## 1. Introduction

The demand for the Global Navigation Satellite System (GNSS) has increased in recent years because many applications use the GNSS receiver as a positioning device. With the advent of the low-cost multi-GNSS and multi-frequency receivers, the use of high-accuracy positioning will be accelerated [1]. The real-time kinematic GNSS (RTK-GNSS) is a popular technique for achieving precise positions at the centimeter level because mobile network operators have been installing GNSS base stations in Japan since 2018 [2]. Furthermore, the Centimeter Level Augmentation Service has officially been implemented through the Quasi-Zenith Satellite System (QZSS) since November 2018 [3]. The RTK-GNSS requires observation data from the base station, which means that it is easy to use provided a GNSS device and mobile network are available. Based on our various test results using a car in Tokyo, the performance of the RTK-GNSS using a low-cost GNSS receiver was found to be relatively good, and the fix rate was always over 80% to 90% in normal urban areas near our university campus. However, it is well known that the fix rate decreases dramatically when moving into dense urban areas surrounded by many high-rise buildings. In reality, the RTK-GNSS is almost impossible to implement on a street surrounded by many high-rise buildings. In addition, RTK-GNSS has been used in surveys for construction in open sky conditions. However, the requirement of GNSS surveys in challenging areas is increasing. For example, landside monitoring under conditions in which the sky is approximately half blocked and surveying nearby high-rise buildings. Automatic parking position management using GNSS is also becoming popular. If the parking location is in dense urban areas, there are numerous NLOS signals. To achieve RTK-GNSS positions and better (Differential GNSS) DGNSS positions, NLOS detection is very important even in the case of static conditions. Therefore, we focused on improving the RTK-GNSS and DGNSS in such a challenging area. The key to improving the performance of the RTK-GNSS is whether satellites contaminated by large multipath errors can be detected, and approximately 1- to 2-m level accuracy is required for reliable carrier phase ambiguity resolution of the RTK-GNSS [4]. Therefore, the use of 3D building models and the fish-eye view have been proposed [5,6,7]. Multipath detection using machine learning has also been presented in recent years [8]. Furthermore, GNSS/Inertial Navigation System (INS) integration has been proposed for continuous positioning and was found to be effective in reducing the large jumps resulting from multipath errors [9]. Although the abovementioned methods are powerful tools for detecting non-line-of-sight (NLOS) signals, the performance of the RTK-GNSS needs to be improved.

The aim of this study was to demonstrate how to detect the satellite of an NLOS signal using only the GNSS. We have previously presented several methods using the carrier to noise ratio (C/N_0_) to reduce multipath errors, one of which is to set the elevation-dependent threshold of C/N_0_ [10]. If C/N_0_ is over this threshold according to the elevation angle, the satellite is used for positioning. This method is relatively effective for detecting NLOS signals. However, the C/N_0_ of the NLOS signal occasionally reaches the same signal level as the usual line-of-sight (LOS) signal. Another method is to use the C/N_0_ fluctuation magnitude for a certain period [11]. The disadvantage of this approach is that it is difficult to set the optimal threshold. In fact, the magnitude of the multipath signal fluctuation in dense urban areas frequently changes from small to large. We generally use the weighted least-squares method to determine the position, which provides the position as well as each residual for the satellites used in positioning. These residuals can be used for consistency checks to detect satellites with large residuals [12]. In terms of GNSS receiver correlator-based techniques, several popular methods are available, such as the narrow correlator, strobe correlator, and multipath parameter estimation [13,14,15]. These techniques are effective in reducing multipath errors provided an LOS signal is received. In the case of receiving an NLOS signal, the correlator occasionally tracks an NLOS signal because the amplitude of an NLOS signal is higher than the amplitude of an attenuated LOS signal.

Our new approach for detecting an NLOS signal is simply based on the continuous time-series C/N_0_ for a certain period. If the local minimum of the satellite C/N_0_ is less than the determined threshold, the satellite is not used for positioning for a certain period, even if the C/N_0_ is recovered over the threshold. The threshold is determined using the average C/N_0_ values according to the elevation angle. The threshold is actually set at −10 dB-Hz from the above-average values in each satellite system. Once the satellite C/N_0_ decreases to less than the determined threshold, these signals are considered to be inappropriate for positioning for a certain period because an LOS signal without strong reflection never decreases below the threshold. The details for determining the period in which the satellite is not used for positioning are introduced later.

The study consisted of the following, as outlined in the remainder of the paper: first, the C/N_0_ values of all GNSS satellites obtained for 30 min in a typical environment that was likely to receive NLOS signals were thoroughly investigated. Simultaneously, the postprocessed pseudo-range residuals for all GNSS satellites were estimated by providing the precise antenna position, and these were compared to identify the features of satellites contaminated by NLOS signals. In this case, the NLOS signal included both a reflected wave and diffracted wave. To estimate the residuals, one QZSS was selected to estimate the receiver clock error correctly because the highest satellite was QZSS 195, and its multipath error was very small, even in dense urban areas. Thereafter, GNSS with a large pseudo-range residual over 15 m was selected to verify the C/N_0_ fluctuations. Based on these data, our new approach for detecting NLOS signals was introduced. We also confirmed the validity of using both the threshold and period of exclusion from positioning for this approach. Subsequently, three test configurations were established, and the performances of the differential GNSS (DGNSS) and RTK-GNSS were evaluated while using the proposed method. In all three tests, the DGNSS positioning accuracy was improved dramatically using the selected satellites based on the new concept. Moreover, numerous correct fixed solutions of the RTK-GNSS were validated. In fact, few correct fixed solutions were obtained without the new satellite selection method.

## 2. Features of Measured C/N_0_ in Dense Urban Areas

The methods to degrade GNSS positioning in dense urban areas are divided into three. First, when signals are completely blocked, they cannot be received and thus degrade the satellite geometry; such signals sometimes make positioning impossible. Second, when the direct signal is blocked (or attenuated) but the signal can be received via a reflected path, the process is called NLOS reception. NLOS signals sometimes exhibit very large errors reaches 100 m or more. The diffracted signal is also included in NLOS signals. Third, when both direct LOS and reflected signals are received, we observe that the errors fluctuate in both the positive and negative directions. This type of multipath error can be somehow mitigated by the signal and receiver designs. Figure 1 presents the three typical multipath types described here.

### 2.1. Measured C/N_0_ in Dense Urban Areas

To investigate the characteristics of C/N_0_ in dense urban areas, we obtained raw measurements near high-rise buildings in Tokyo. The configuration of the target location is illustrated in Figure 2. The right side of the figure shows the 3D Google map, and the red circle represents the GNSS antenna location. The left side of the figure shows the 2D Google map and the height information of the surrounding buildings at this location. In this figure, the top denotes the north. As can be observed from Figure 2, it was likely that NLOS signals would be received from the higher buildings on the azimuth of 290°. Data from the GPS, GLONASS, BDS, QZSS, and Galileo were recorded on a street near Tokyo Station in a typical urban-canyon area for 30 min from 8:15:00 GPS time (GPST) on April 3, 2020. A u-blox F9P, which is a low-cost dual-frequency receiver with a standard patch antenna, was used to record the data at 1 Hz. These data were also used to verify the pseudo-range residuals. Figure 3 presents the sky plots of all satellites over a 15° mask angle at 8:25:00 GPST and the condition of the satellites. A 3D map was also used in this case to specify the multipath types of all satellites. The satellite position and estimation of the radio wave propagation path were calculated using previously developed software known as GPS-Studio [16]. In this study, the accuracy of the 3D map data GEOSPACE3D was approximately 1.5 m, and the map did not include trees and signs. The building shapes in the study area were nonuniform in the height direction. Hence, the building shape was modified for this study using GIS software called QGIS to prevent incorrect estimation of the propagation path. Modifications were made within a circle with a radius of 500 m that was centered on the antenna under evaluation. In Figure 3, the white and black circles indicate the blocked or NLOS satellites, whereas the red circles indicate the LOS satellites. The four satellites GPS 01, QZSS 03, Galileo 01, and BDS 07 were selected to demonstrate the temporal C/N_0_ of the L1 band for 30 min as LOS signals. We selected these four satellites because these they were clearly regarded to receive LOS signals all the time based on a 3D map. Moreover, the four satellites GPS 22, GPS 27, GPS 30, and GLONASS 02 were selected to demonstrate the temporal C/N_0_ of the L1 band for 30 min as NLOS signals because these four satellites were clearly regarded to receive NLOS signals all the time based on a 3D map. Figure 4 shows the temporal C/N_0_ mainly receiving LOS signals. Figure 5 shows the temporal C/N_0_ mainly receiving NLOS signals. TOW represents time of week in GPST. As can be observed from Figure 4, the C/N_0_ of the LOS signals did not fluctuate substantially, with the exception of Galileo 01. The C/N_0_ value of Galileo 01 was first affected by diffraction and then received LOS + reflected signal for a certain time. As can be observed from Figure 5, the C/N_0_ of the NLOS signals fluctuated strongly in all satellites. In fact, other satellites that recognized NLOS reception, which are not shown in Figure 5, exhibited a similar tendency. The LOS signals of GPS 22, GPS 27, and GLONASS 02 were clearly blocked by the buildings on the azimuth of 110°. However, the C/N_0_ values of the three satellites were mostly sufficient to track the carrier phase measurements over approximately 30 dB-Hz. More importantly, stable C/N_0_ could not be observed owing to the specular reflection, despite these signals of the three satellites clearly being reflected by the tall building on the azimuth of 290°. This indicates that the diffracted signal owing to the building on the azimuth of 110° could be received with NLOS signals (reflected by the building on the azimuth of 290°) simultaneously. This phenomenon is also illustrated in case 2 in Figure 1. Regarding the C/N_0_ of GPS 30, the signal was often completely lost owing to the tall building between GPS 30 and the antenna, which is exactly the situation of case 1 in Figure 1. 

### 2.2. Estimation of Pseudo-Range Errors

This section describes the method used to estimate the errors between the measured and predicted geometrical ranges. The estimated accurate errors were significant in evaluating whether our proposed method using C/N_0_ was valid for detecting multipath-contaminated satellites. Pseudo-range measurements include error sources from the receiver clock, satellite clock, ephemeris, ionosphere, troposphere, and multipath + noise. In this study, the predicted geometrical range was already set as the distance between the accurate positions of the antenna and the satellite. All above error sources, with the exception of the receiver clock and multipath + noise, can be modeled within several meters in total. Hence, a method for estimating the receiver clock error was required. In challenging areas, receiver clock errors, which are similar to multipath errors, occasionally jump and deviate over tens of meters. When the receiver clock error is determined within several meters, each residual can be precisely predicted, and these residuals are almost equivalent to multipath errors. In this study, we could track a good signal from the Japanese QZSS even near high-rise buildings in Tokyo as at least one QZSS remained at a very high elevation angle over 80°. In reality, the accuracy of the pseudo-range of the highest QZSS was within 1.0 m. The summary and flow of the above method are presented in Figure 6, including equations. Note that it is important to estimate the bias between the GPS and other satellite systems, which can be achieved at the base station.

According to the flow presented in Figure 6, we estimated the receiver clock error using a satellite, which was the highest elevation. At this time, QZSS 03 was the satellite with the highest elevation. These estimated receiver clock errors were applied for the other satellites, thereby generating an estimate of the errors for each satellite within several meters. This connotes that, if satellites are selected within a certain threshold of these errors, the accuracy of the DGNSS will be improved from tens of meters to several meters. To confirm the accuracy of the above estimated errors, the accuracy of the DGNSS was evaluated at the same location, with 30 min of data analyzed. The base station is on the rooftop of our laboratory, and the baseline length is 2.6 km. As the accuracy of the deduced pseudo-range error was approximately several meters, the threshold for selecting the satellites was set to 10 m. If the estimated pseudo-range error for each satellite was more than 10 m, the satellite was not used for DGNSS. Therefore, the horizontal standard deviation of the DGNSS was 2.8 m, and the average horizontal position was only 0.9 m when using the selected satellites. However, in the case of the normal DGNSS, the horizontal standard deviation was 16.7 m and the average horizontal position was 40.3 m without the information of the selected satellites. The above result indicates that the proposed residual estimation is clearly effective for distinguishing large multipath errors such as those of more than 10 m in dense urban areas.

### 2.3. Relationship Between Pseudo-Range Errors and C/N_0_

This section demonstrates the relationship between the pseudo-range error and C/N_0_ for all satellites with an elevation angle over 15°. Based on the above method for estimating the pseudo-range error, we analyzed the 30 min of GNSS data used previously. The results are presented in Table 1. For each satellite vehicle (SV), the condition of satellite, the maximum pseudo-range error, 90th percentile of the pseudo-range error, 10th percentile of the pseudo-range error, percentage of this time to the total time that C/N_0_ was less than 30 dB-Hz, percentage of this time to the total time that C/N_0_ was less than 40 dB-Hz and more than 30 dB-Hz, and percentage of this time to the total time that C/N_0_ was more than 40 dB-Hz are shown in Table 1. As for the condition of each satellite, when the percentage of the LOS condition was 90% or more for 30 min, the satellite was regarded as LOS. When the percentage of NLOS condition was 90% or more for 30 min, the satellite was regarded as NLOS. When the percentage of LOS or NLOS condition was less than 90% for 30 min, the satellite was regarded as partial. The analysis conditions for each satellite were as follows: the elevation mask angle was 15°, and both the pseudo-range and carrier phase measurements had to be generated. The cycle slip and half-cycle validation for the carrier phase measurement were not checked. Note that not all satellites were constantly output for 30 min because the signals of certain satellites were completely blocked by buildings. Therefore, the sum of the percentages for C/N_0_ was not always 100%. When calculating the maximum error and percentile, the error was converted into the absolute error. The error for QZSS 03 was quite small because QZSS 03 was used as the reference satellite for estimating the receiver clock error. Important clarifications regarding LOS or NLOS for each satellite were obtained using a 3D map and precise antenna position, as previously mentioned in Section 2.1.

The pseudo-range error of GPS 22 was over 50 m in more than 90% of the 30 min. The percentage of this time to the total time that C/N_0_ was more than 40 dB-Hz was higher than 67%. The percentage of this time to the total time that C/N_0_ was more than 30 dB-Hz was 95%. Even if the signal of GPS 22 was clearly affected by the NLOS signal, the C/N_0_ of GPS 22 was maintained at a sufficiently high level to track the carrier phase measurements, as illustrated in Figure 5. This indicates that it is very difficult to detect satellites affected by strong NLOS signals by using the normal C/N_0_ threshold. We usually set the minimum threshold for C/N_0_ at approximately 30 to 35 dB-Hz. Even if we set 40 dB-Hz as a threshold in this case, GPS 22 with a large pseudo-range error was included in the positioning for 67% of the total duration. This is exactly the case we need to consider to improve the accuracy in dense urban areas. Similar trends to GPS 22 as described above could be observed for other satellites, such as GPS 03, GPS 08, GPS 27, Galileo 21, and GLONASS 02. However, the pseudo-range error of GPS 01 was less than 5 m for 30 min. The percentage of this time to the total time that C/N_0_ was more than 40 dB-Hz was 100%. Similar trends to GPS 01 as described above could be seen for other satellites, such as GPS 03, GPS 11, QZSS 02, QZSS 03, QZSS 07, Galileo 01, BDS 07, BDS 08, BDS 25, and GLONASS 15. We noticed that the C/N_0_ of these satellites did not decrease by 30 dB-Hz and C/N_0_ did not fluctuate substantially. The C/N_0_ values of the satellites receiving LOS signals fluctuated occasionally because of the reception of the reflected signals, as indicated in Figure 1. In such cases, the correlator-based multipath mitigation technique is very powerful for reducing multipath errors, as described in Section 1 [13,14]. According to these results, the C/N_0_ of satellites affected by strong NLOS signals can easily reach 40 dB-Hz and more while exhibiting decreases below 30 to 35 dB-Hz. Furthermore, the larger pseudo-range errors of these satellites continued for a long time, even though large fluctuations could be observed. 

## 3. Proposed Strong Multipath Detection Using C/N_0_ Information

In this section, we introduce a new concept for detecting satellites affected by strong multipath, especially in the case of NLOS signal reception. As demonstrated in Section 2.3, we identified two important facts regarding the relationship between the pseudo-range error and C/N_0_ in the case of receiving NLOS signals, which are as follows:The C/N_0_ of satellites affected by strong NLOS signals can easily reach 40 dB-Hz and above. They also fluctuate significantly from below 30 dB-Hz to over 40 dB-Hz. Furthermore, the large pseudo-range errors of these satellites continue for a long time even though large fluctuations can be observed.The C/N_0_ values of satellites that clearly receive LOS signals do not decrease below 30 dB-Hz, and relatively high C/N_0_ values of more than 40 dB-Hz are maintained. Moreover, they sometimes fluctuate owing to the reception of the reflected signal. However, these pseudo-range errors can be reduced using correlator-based multipath mitigation because they receive LOS signals. The pseudo-range errors are mostly less than 5 m.

Based on these two facts, it is very difficult to detect NLOS signals by setting the minimum threshold for the C/N_0_ of satellites because high C/N_0_ values of more than 40 dB-Hz can easily be reached. Therefore, we focused on determining whether the C/N_0_ value decreases below the threshold. When the value of the satellite C/N_0_ is less than the threshold, the satellite is not used for positioning for a certain period. Figure 7 depicts our new method for detecting satellites with large pseudo-range errors using C/N_0_ information. It illustrates the temporal C/N_0_ for GPS 03 in the dense urban areas depicted in Section 2.1. Two important parameters needed to be set for this method: the C/N_0_ threshold and the period in which the satellite was not used for positioning. As an example, in Figure 7, we set 30 dB-Hz and 1 min as the threshold and period, respectively. Those values are just examples. As indicated in the figure, even if the value of C/N_0_ recovered over 30 dB-Hz after 462,460 s, the satellite was not used for positioning for at least 1 min. After 462,520 s, the satellite could be used for positioning but it returned to being unusable for positioning after only 24 s. In reality, GPS 03 could not be used for positioning from 462,520 to 462,544 s because the pseudo-range error was continuously large over 60 m.

### 3.1. Method to Determine the Threshold and Period

We explain how to set both the threshold and period properly. Regarding the threshold, we consider that the elevation-dependent C/N_0_ for each GNSS obtained in the open sky conditions is suitable because C/N_0_ generally changes according to the elevation angle. For most GNSS satellites, the C/N_0_ is low at a low elevation angle and high at a medium or high elevation angle. The difference between the lowest and highest C/N_0_ for each GNSS is more than 10 dB-Hz, except for the case of the geostationary satellite (GEO). Figure 8 presents the test results of C/N_0_ for GPS L1-C/A according to the elevation angle obtained for 24 h in the open sky condition. The receiver used in this test was the u-blox F9P, as in the test described in Section 2.1. The mask angle for the elevation was set to 15°. All observed C/N_0_ values are indicated by black plots. The average C/N_0_ depending on the elevation angle is depicted by the gray line. The minimum C/N_0_ obtained in this test was 32 dB-Hz when the elevation angle was less than 20°. The maximum C/N_0_ obtained was 50 dB-Hz when the elevation angle was more than 50°. The gray line indicates the average signal strength according to the elevation angle for GPS L1-C/A. If C/N_0_ of the satellite decreased by more than approximately 10 dB-Hz from this average C/N_0_, the satellites were clearly considered to exhibit diffraction or reflection received by the antenna based on the previous test results presented in Table 1. The black dotted line indicates our proposed threshold of C/N_0_, which was set to 10 dB-Hz lower than the average C/N_0_ depending on the elevation angle. Figure 8 depicts the case of GPS L1-C/A. We prepared each threshold for the other GNSS in the same manner. The method of deciding the suitable threshold is discussed as follows. As can be seen in Figure 8, if we set 5–6 dB-Hz below from the average line, we might remove a favorable satellite by mistake although it has a very small multipath error within 1 m, which should not be done. This is based on measurement but can be applied in any case. Conversely, if we set 14–15 dB-Hz below from the average line, we sometimes cannot detect the NLOS signal under challenging areas. Considering at the temporal C/N_0_ of GPS 22 and GLONASS 02 shown in Figure 5, we cannot detect these satellites of NLOS for a long time. Although this fact is based on the measurement, this threshold of 14-15 dB-Hz is not suitable for our proposed method. Therefore, we set the threshold from approximately 6-14 dB-Hz. Even with the empirical data, we can decide the range of the threshold although we cannot decide the best threshold. In addition, regarding the several types of GNSS receivers, we investigated the consistency of C/N_0_ under open sky conditions using 24-h data, and the difference between receivers was very small within 1-2 dB-Hz as long as we used the antenna and cable recommended by manufacturers. Furthermore, we showed the test results at three different locations showing how the threshold choice influences the test results in Section 5.

For the setting of the period that should not be used for positioning, we investigated all satellites with large pseudo-range errors, as indicated in Table 1. In particular, satellites with pseudo-range errors over 15 m in the 90th percentile of all errors were selected for investigation. If the C/N_0_ was continuously over the threshold determined above, we counted the number of seconds. Table 2 summarizes the longest period, second longest period, and third longest period for each selected satellite. It also includes information regarding the 90th percentile of all errors and the percentage of this time to the total time that C/N_0_ was more than 40 dB-Hz. For Galileo 31, the pseudo-range error was small for most of the time over 80% of the total. Therefore, no information was available for the second and third longest periods. For GLONASS 16, large pseudo-range errors occurred in the final 30% of the duration when C/N_0_ decreased below 30 dB-Hz. For the remaining 70% of the duration, C/N_0_ was high and the pseudo-range errors were small. Therefore, no information was available regarding the continuous period over the threshold. As can be observed from Table 2, the longest period in total was 164 s for GPS 22, and the second longest period in total was 163 s for GLONASS 02. These two satellites were affected by strong NLOS signals from the tall building on the azimuth of 290°, as illustrated in Figure 2 and Figure 3. Figure 9 presents the case of the temporal C/N_0_ for GPS 22 with the elevation-dependent threshold, in which the three longest periods according to both C/N_0_ and the proposed threshold are depicted. Furthermore, GPS 08 and Galileo 21 were also affected by strong NLOS signals from the tall building on the azimuth of 290°. Therefore, the longest periods for both satellites were over 80 s. Conversely, the C/N_0_ of the satellites that received continuous strong NLOS signals decreased below the threshold determined according to the above at least approximately every 160 s. The main reason for this is that the diffracted signal from the building on the azimuth of 110° was actually received to the antenna simultaneously. Another possible explanation is that the reflectance of the wall of the building on the azimuth of 290° changed according to the reflected point. Based on these results presented in Table 2, if we set the period that should not be used for positioning to approximately 180 s, we could mostly remove these satellites with large pseudo-range errors from the positioning in the case of these data. However, the suitable period will change according to several factors, such as the distance between the antenna and surrounding buildings, the reflection, and diffraction degrees. Therefore, we need to set the period in which the satellite is not used for positioning to at least 180 s. For example, if the distance between the antenna and building reduces, the period of fluctuation owing to the multipath becomes long. Examining this test configuration, it is clear that the distances between buildings are large. Therefore, according to the test configuration, the period should be set to much longer than 180 s. In this study, we used 240 s as the period and compared the results when changing the period to 180 and 300 s. Moreover, the remaining large pseudo-range error could be detected using the normal residual check in the least-squares method. With this proposed method, it is important to balance the number of available satellites for positioning because at least four satellites are required for positioning. With the advent of multiple GNSS in recent years, numerous satellites are available even in dense urban areas. Furthermore, low-cost multi-frequency and multi-GNSS receivers have emerged. Therefore, we can focus on removing the satellites with large pseudo-range errors from positioning without significant hesitation.

### 3.2. Flowchart of the Proposed Method

Figure 10 depicts the flowchart of our proposed method, where our new approach is indicated in the gray-colored part. The detection method of LOS or NLOS is detailed in this part. Positioning software developed by our laboratory was used. The core part of the program was considerably similar to RTKLIB, and we modified certain functions such as the satellite selection to include the new concept. We also used velocity information deduced from the Doppler frequency [17]. RTKLIB is an open-source program package for standard and precise positioning with the GNSS [18]. First, GNSS observation data from both the base station and rover station were input. Thereafter, several satellite selection methods were performed. Both the pseudo-range and carrier phase measurements had to be generated from the receiver. Furthermore, half-cycle ambiguities had to be resolved in the case of the RTK-GNSS. The flag for this half-cycle checking from the receiver could be used for this purpose. The normal mask angle and required minimum C/N_0_ were set for both the base station and rover station. The mask angle was set to 15°. The value of the minimum C/N_0_ for the L1 band (GPS and QZSS L1-C/A, GLONASS G1, Galileo E1, and BDS B1) was set to 32 dB-Hz, which is actually the minimum C/N_0_ for all GNSS obtained in the open sky condition when the mask angle is set to 15°. Furthermore, the value of the minimum C/N_0_ for the L2 band (GPS and QZSS L2C, GLONASS G2, Galileo E5B, and BDS B2) was also set because we could expect multi-frequency diversity [19]. As the wavelength differed for the L1 band and L2 band frequencies of the same satellite system, the fluctuation period of C/N_0_ owing to diffraction or reflection was different for the L1 band and L2 band. The minimum C/N_0_ for L2P of the GPS was not checked because the u-blox F9P receiver does not output L2P signals. For the rover station, after applying these normal settings as in the base station, the time-series C/N_0_ values were checked in terms of the threshold and period based on our proposed method. Subsequently, the residuals of the satellites in the least-squares method were checked. If the absolute residual of the satellite was the maximum and over 10 m, the satellite was repeatedly removed from the positioning provided that the HDOP was lower than 10. Finally, we processed these measurements to obtain the DGNSS and RTK-GNSS solutions. We compared the performance with or without checking the time-series C/N_0_ based on our new approach. The other settings mentioned in the above remained exactly the same.

### 3.3. Other Test for the Generalization of the Data

To evaluate the presented method, 30 min data were used in Section 2 and Section 3. To give a strict justification for the generalization of the data, we tested another two datasets of 1 h. One dataset was obtained from near many high-rise buildings that were likely to receive NLOS signals in Tokyo on 6 July 2020. The distance between buildings was approximately 37 m. Another dataset was obtained from near medium-rise buildings that were also likely to receive NLOS signals in Tokyo on 7 July 2020. The distance between buildings was approximately 15 m. As we mentioned regarding the proposed method, two parameters are important to detect the large pseudo-range errors. Based on the above new data sets, we verified if our presented threshold and period was suitable or not although we could not decide the best parameters. Similar to Section 3.1, satellites with pseudo-range errors over 15 m in the 90th percentile of all errors were selected for investigation. Figure 11 shows the longest period of selected satellites for three different thresholds, 8, 10, and 12 dB-Hz, in the case of high-rise buildings. Figure 12 shows the longest period of selected satellites with three different thresholds, 8, 10, and 12 dB-Hz, in the case of medium-rise buildings. It is natural that the lower the threshold, the shorter the longest period. However, if we set a lower threshold, the number of used satellites in positioning will reduce. Thus, these settings have to be balanced. As can be seen in Figure 11, most satellites can be removed from positioning if we set 4 min as a period in the case of 10 dB-Hz as a threshold. However, GPS 11 and Galileo 03 cannot be removed from positioning because the longest period was over 300 s in the case of 10 dB-Hz as a threshold. Considering 8 dB-Hz as a threshold, GPS 11 can be removed but Galileo 03 is still left. If we set 4 min as a period and 8 dB-Hz as a threshold, the large pseudo-range error of Galileo 03 will be left for about 20 s. In reality, because pseudo-range residual checks are conducted after LOS/NLOS detection, as shown in Figure 10, the satellites that have large pseudo-range errors will be removed eventually if the number of satellites that have large pseudo-range errors is less. As can be seen from Figure 12, all satellites will be removed from positioning if we set 4 min as a period in the case of 10 dB-Hz as the threshold. From these two test results, although one or two satellites might be left after our new detection method of NLOS signals, most satellites can be removed from the positioning. In addition, pseudo-range residual checks after the new detection method will be effective to remove one or two satellites that have large pseudo-range errors because the residual check is basically effective when the pseudo-range error is large for only one satellite.

Using these two datasets, we also verified the characteristics of NLOS signals in the relatively challenging areas, such as those mentioned at the beginning of Section 3. The C/N_0_ of satellites affected by strong NLOS signals can easily reach 40 dB-Hz and above. In addition, the large pseudo-range errors of these satellites continue for a long time even though large fluctuations can be observed as shown in several satellites both in Figure 11 and Figure 12. The C/N_0_ values of satellites that clearly receive LOS signals do not decrease below 30 dB-Hz, and relatively high C/N_0_ values of more than 40 dB-Hz are maintained.

## 4. Testing and Results

In this section, the three urban scenarios selected for the experiments to evaluate our proposed method are described, and the results are discussed in detail. The first scenario was a dense urban area close to the previous test location described in Section 2.1. The characteristic of this location was that several NLOS signals were always received from tall buildings on one side. The second scenario was also a dense urban area surrounded by many high-rise buildings, with fewer LOS signals than in the first scenario. The third scenario was a location surrounded by two medium-rise buildings on both sides on our campus. The characteristic of this location was that several NLOS signals were always received from medium-rise buildings on one side. The raw data of the GNSS were post-processed using our algorithm mentioned above. The processing is rapid and can be used in real time. The base station on the rooftop of our building was used for correction data. The baseline length was less than 5 km in each test.

### 4.1. Test Results in the First Location

The experiment was performed in a dense urban environment near Tokyo station in Tokyo, Japan, on May 27, 2020. The configuration of the receiver and antenna used in all three tests is presented in Table 3. The car was parked at the edge of the road and a standard patch antenna was attached to the car rooftop. The sampling rate of the measurement data was 1 Hz. The satellite systems and frequencies used in this test were GPS/QZSS L1C/A and L2C, Galileo E1 and E5B, GLONASS G1 and G2, and BDS B1 and B2. The L2P signals of the GPS and GEO satellites of BDS were not used because they were not available in this GNSS receiver. The total data recording period was 1800 s. Postprocessed RTK positioning was used to estimate the precise reference position. As mentioned in Section 3, the positioning software used in this study was developed in our laboratory. The same positioning software was used for all three tests. The common settings without our proposed method for the satellite selection are listed again in Table 4. The residuals in the least-squares method were checked repeatedly provided that the absolute residual was over 10 m. The test location was very close to the case of Section 2.1 illustrated in Figure 2. Moreover, five images around the car are presented in Figure 13 to provide a better understanding of the situation around the antenna. There were approximately three to four lanes on each side of the road and several high-rise buildings with different heights on both sides. Several trees and a large iron signboard over the car were also present. 

The horizontal errors of the DGNSS are depicted in Figure 14. The horizontal errors of the DGNSS were occasionally reduced by the residual checks; however, very large errors remained. The maximum absolute horizontal error in the DGNSS was 91.13 m, and the 90th percentile value was obtained as 37.80 m. The standard deviations in the horizontal direction for E and N were 16.75 and 6.13 m, respectively. The average errors in the horizontal direction for E and N were −8.49 and −2.11 m, respectively. The accuracy was not improved in the case in which different minimum C/N_0_ values were set, namely 35 and 40 dB-Hz. In reality, no correct fixes of the RTK-GNSS occurred, although the fix rate was 12.4%. The fix rate was calculated by dividing the total epochs by the number of fixed solutions. If the absolute horizontal error was less than 10 cm and the absolute height error was less than 20 cm from the true position, the solution was regarded as the correct fix. It was difficult for the RTK-GNSS to resolve correct ambiguities because the absolute position deduced from the pseudo-range based on the DGNSS was mostly deviated and unstable. 

The horizontal errors of the DGNSS using our new method are presented in Figure 15. Note that the vertical scale is different from that of Figure 14. The absolute horizontal error was reduced substantially from the beginning to the end, indicating that the selection of multipath-contaminated satellites was quite effective when using our new method and the unfavorable satellites could quickly be removed in order. The maximum horizontal error was 7.85 m, and a 90th percentile value of only 4.31 m was obtained. The standard deviations in the horizontal direction for E and N were 2.02 and 1.06 m, respectively. The average errors in the horizontal direction for E and N were 1.24 and −0.67 m, respectively. The accuracy was improved dramatically using the new satellite selection method. Regarding the RTK-GNSS, the fix rate was 47.2% and all solutions were correctly fixed. The standard deviations of the RTK-GNSS in the horizontal direction for E and N were 0.015 and 0.017 m, respectively.

### 4.2. Test Results in The Second Location

The experiment was performed in a dense urban environment near Tokyo station in Tokyo, Japan, on April 18, 2020. The configuration of the receiver and antenna used in this test was the same as that described in Section 4.1. The car was parked at the edge of the road, and the standard patch antenna was attached to the car rooftop. The sampling rate of the measurement data was 1 Hz. The satellite systems and frequencies used in this test were also the same as those described in Section 2.1. The total data recording period was 1,800 s. Post-processed RTK positioning was used to estimate the precise reference position. The configuration of the target location is presented in Figure 16. The left side of the figure depicts the 3D Google map, and the circle indicates the GNSS antenna location. The right side of the figure presents images around the car. There were three lanes on each side of the road, and several high-rise buildings with different heights were present on both sides. 

The horizontal errors of the DGNSS are presented in Figure 17. The tendency of the errors was similar to that of the previous test results, although the maximum error was substantially larger. The maximum absolute horizontal error in the DGNSS was 175.50 m, and the 90th percentile value obtained was 31.55 m. The standard deviations in the horizontal direction for E and N were 21.95 and 15.66 m, respectively. The average errors in the horizontal direction for E and N were 6.01 and 5.88 m, respectively. The accuracy was not improved in the case in which different minimum C/N_0_ values were set, namely 35 and 40 dB-Hz. In reality, the fix rate was 16.5%, and the correct fixes out of the total epochs were only 9.8%. The standard deviations of the RTK-GNSS with correct ambiguities in the horizontal direction for E and N were 0.013 and 0.012 m, respectively.

The horizontal errors of the DGNSS using our new method are depicted in Figure 18. Note that the vertical scale is different from that of Figure 15. The absolute horizontal error was reduced substantially from the beginning to the end. This indicates that the selection of multipath-contaminated satellites was quite effective when using our new method, and the unfavorable satellites could be rapidly removed in order. The maximum horizontal error was 31.53 m, and the 90th percentile value obtained was only 9.84 m. The standard deviations in the horizontal direction for E and N were 3.00 and 5.10 m, respectively. The average errors in the horizontal direction for E and N were 0.57 and 2.04 m, respectively. The accuracy was dramatically improved when using the new satellite selection method. Regarding the RTK-GNSS, the fix rate was 26.4% and the correct fixes out of the total epochs totaled 23.0%. The number of incorrect fixes was 3.4%. The standard deviations of the RTK-GNSS with correct ambiguities in the horizontal direction for E and N were 0.012 and 0.018 m, respectively.

### 4.3. Test Results in Third Location

The experiment was performed at our Etchujima campus in Tokyo, Japan, on 17 April 2020. The configuration of the receiver and antenna used in this test was same as that described in Section 4.1. The car was parked near a two-story building, and the standard patch antenna was attached to the car rooftop. There was a seven-story building on the other side. Therefore, it was likely that reflected NLOS signals from the side of the seven-story building would be received. Furthermore, diffracted NLOS signals were often received from the side of the two-story building. The main difference between the first two tests and this test was that the distance from the buildings was short and the buildings were not very tall. The sampling rate of the measurement data was 1 Hz. The satellite systems and frequencies used in this test were also the same as those described in Section 2.1. The total data recording period was 1800 s. The postprocessed RTK positioning was used to estimate the precise reference position. The configuration of the target location is depicted in Figure 19. 

The horizontal errors of the DGNSS are depicted in Figure 20. The standard deviations in the horizontal direction for E and N were 1.62 and 1.87 m, respectively. The average errors in the horizontal direction for E and N were 11.76 and −10.37 m, respectively. The maximum absolute horizontal error in the DGNSS was 22.45 m, and the 90th percentile value obtained was 18.43 m. The accuracy was not substantially improved in the case where different minimum C/N_0_ values were set, namely 35 and 40 dB-Hz. In reality, the fix rate was 44.4%, but the correct fixes out of the total epochs totaled only 7.5%. It was difficult for the RTK-GNSS to resolve correct ambiguities and to maintain the corrected ambiguities because the absolute position deduced from the pseudo-range based on the DGNSS was always deviated.

The horizontal errors of the DGNSS using our new method are illustrated in Figure 21. The deviations of the horizontal error were reduced significantly from the beginning to the end, although several jumps could be observed. This indicates that the selection of multipath-contaminated satellites was effective when using our new method, particularly in mitigating NLOS signals. The maximum horizontal error was 22.33 m, and the 90th percentile value obtained was only 4.89 m. The standard deviations in the horizontal direction for E and N were 2.62 and 2.45 m, respectively. The average errors in the horizontal direction for E and N were 2.32 and −1.55 m, respectively. The accuracy in terms of the deviation was improved using the new satellite selection method. Regarding the RTK-GNSS, the fix rate was 97.9%, and the correct fixes out of the total epochs totaled 97.1%. The standard deviations of the RTK-GNSS with correct ambiguities in the horizontal direction for E and N were 0.012 and 0.013 m, respectively.

## 5. Discussion

In this section, the results of the three tests are summarized and briefly discussed. Figure 22 presents a comparison of the RMS errors in the horizontal direction in the three locations for the DGNSS. Figure 23 compares the correct fix rates in the three locations for the RTK-GNSS. The approximate distances between the buildings on either side of the road in the three locations were 43, 41, and 12 m in the order of the experiments. As shown in these two figures, our new multipath detection method was very effective in improving the DGNSS as well as the RTK-GNSS. As depicted in Figure 10, the residuals of the satellites in the least-squares method were checked. If the absolute residual of the satellite was the maximum and was over 10 m, the satellite was repeatedly removed from the positioning. Without this step, the performance of the normal DGNSS was substantially worse than the results shown in Figure 14 and Figure 17 in the case of the relatively long distance between the buildings on either side. In reality, the maximum number of residual checks in a single epoch was 8 in both locations 1 and 2. Errors over 50 m continued for a long time without the residual check step. In this sense, the residual check in the least-squares method was effective in challenging environments surrounded by many high-rise buildings. However, a limitation existed in removing satellites with large pseudo-range errors because large errors still remained, and the method was not effective in the third location because the errors owing to the NLOS signals were not large. For the case of the new approach in the three different locations, the number of residual checks in a single epoch was almost 0. As the large errors were mostly reduced following the proposed time-series C/N_0_ check, the residual check was not as necessary. Even if satellites with large errors remained after using the new approach, it was easy to detect the satellites because the residual check works very effectively if only one or two satellites exhibit large errors. According to the RTK-GNSS results in Figure 23, the correct fix rate was significantly improved at all locations. This is because the RTK-GNSS requires the accuracy of the absolute position based on the pseudo-ranges to be within several meters. If the absolute position deviates by over 10 m, it is impossible to resolve correct ambiguities. 

Figure 24 compares the RMS errors in the horizontal direction in the three locations for the DGNSS using the new method. The results using three different periods were also compared. As discussed in Section 3, we had to set the periods during which the satellite was not used for positioning after the C/N_0_ decreased below the threshold. The lengths of the period were set to 180, 240, and 300 s. As can be observed from Figure 24, the difference among the three periods was not substantial. We assert that a longer period will be required to remove the satellite properly in the case of a short delay of the multipath to the direct path. As illustrated in the case of 12 m between the buildings on each side, a longer period enables the accuracy of the DGNSS to be improved. Figure 25 compares the RMS errors in the horizontal direction at the three locations for the DGNSS using the new method. The results using five different thresholds were also compared. The period was set 240 s. As discussed in Section 3, we had to set the threshold to detect which satellite was not used for positioning. Five thresholds were set 6, 8, 10, 12, and 14 dB-Hz, respectively. As can be observed from Figure 25, RMS errors in all cases were less than the normal DGNSS. Although the best threshold cannot be decided by using these results alone, it seems to be that the favorable range of the threshold was from 8 to 1 dB-Hz. In the case of 12 m between the buildings on each side, the strict threshold setting was allowed. In reality, when we set 20 dB-Hz at three locations, the RMS errors were not improved compared with the normal DGNSS because no satellites were detected by using new method. Regarding the percentage of positioning to total epochs, they were approximately 97% in two locations (1 and 2) when we set 6 dB-Hz as a threshold. This is simply because the strict setting of threshold will decrease the number of used satellites. As for other results, the percentage of positioning to total epochs was 100%.

## 6. Conclusions

This study demonstrated the novel utilization of the C/N_0_ of the GNSS to detect NLOS signals efficiently. Although C/N_0_ values have been used to select GNSS observations with improved quality for many years, most of the previous concepts were based on methods whereby satellites with signal strength over the C/N_0_ threshold were not removed from positioning. In reality, the elevation-dependent threshold of the C/N_0_ setting is somehow effective in mitigating multipath errors. However, the fluctuations in C/N_0_ affected by NLOS signals are quite large. If the C/N_0_ is above the threshold, the satellite is used for positioning even if it is still affected by the NLOS signal, which causes the positioning error to jump easily over several tens of meters. To overcome this issue, we focused on the values of the continuous time-series C/N_0_ for certain periods. If the C/N_0_ of the satellite is less than the determined threshold, the satellite is not used for positioning for a certain period, even if the C/N_0_ recovers over the determined threshold. This new concept is very simple to implement and does not require other equipment or sensors. There are two key parameters for this new method: the determined threshold and the period during which the satellite is not used for positioning. We determined the suitable values for the two parameters based on the test data, which were subsequently evaluated.

Three static tests were conducted at three different challenging locations near high-rise or medium-rise buildings in Tokyo. The test results demonstrated that our new approach could mitigate multipath errors from over several tens of meters to several meters in the DGNSS by detecting NLOS signals. Therefore, the performance of the RTK-GNSS was also improved. Moreover, the standard residual check in the least-squares method was found to be effective in reducing large pseudo-range errors with our new method.

## Figures and Tables

**Figure 1 sensors-20-04059-f001:**
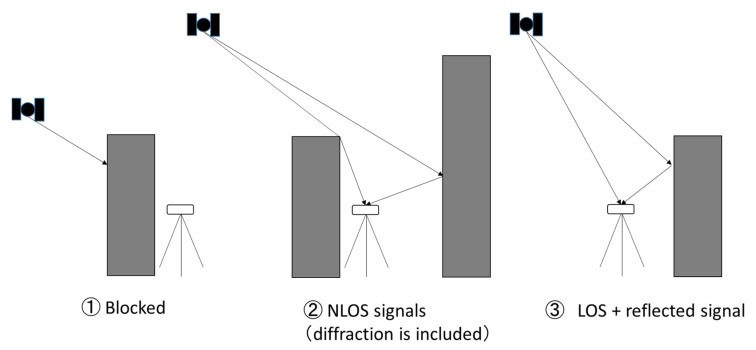
Three multipath types in Global Navigation Satellite System (GNSS).

**Figure 2 sensors-20-04059-f002:**
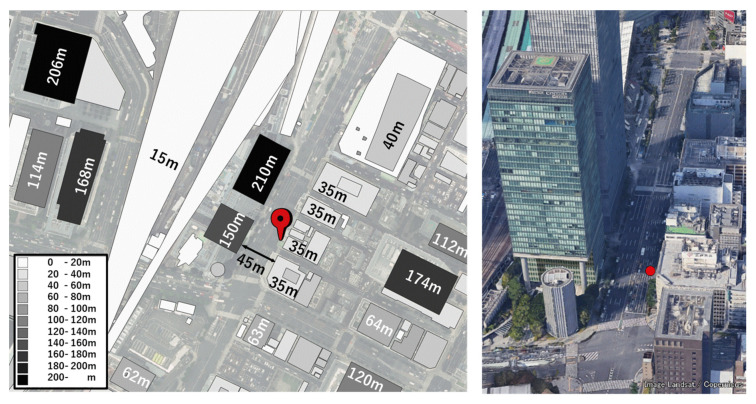
Target location for investigating carrier to noise ratio (C/N_0_) in dense urban areas.

**Figure 3 sensors-20-04059-f003:**
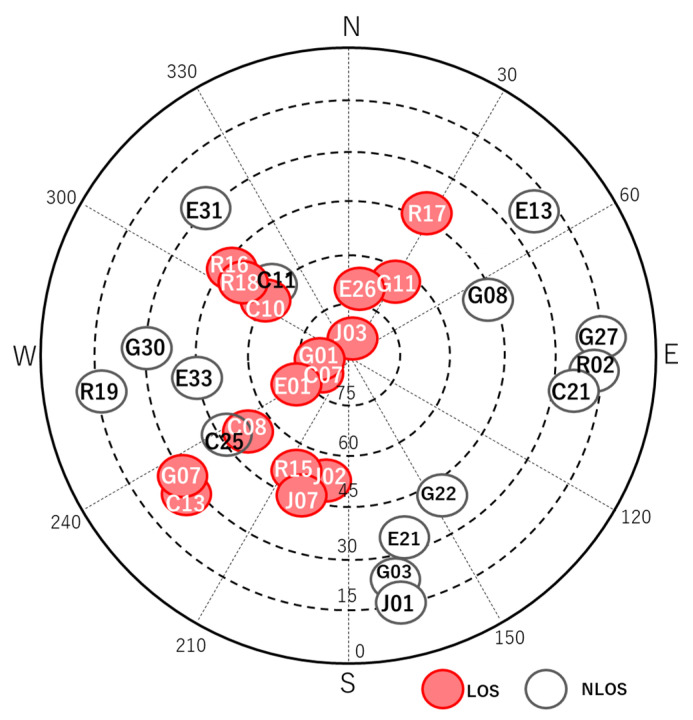
Sky plots of all satellites and their conditions (8:25:00, 462,300 s).

**Figure 4 sensors-20-04059-f004:**
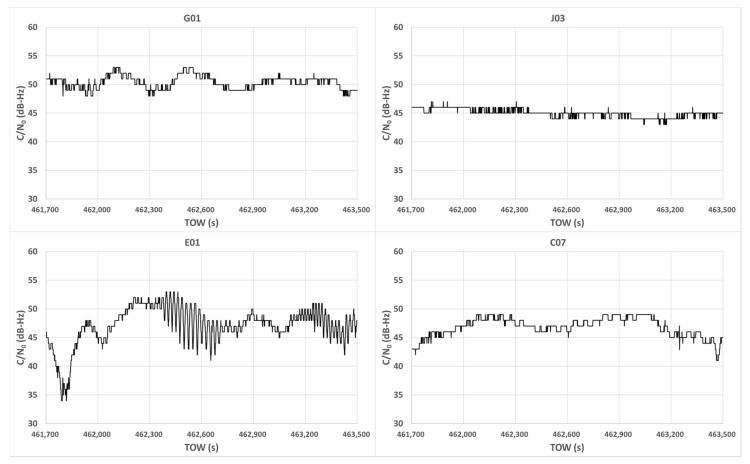
Temporal C/N_0_ mainly receiving line-of-sight (LOS) signals.

**Figure 5 sensors-20-04059-f005:**
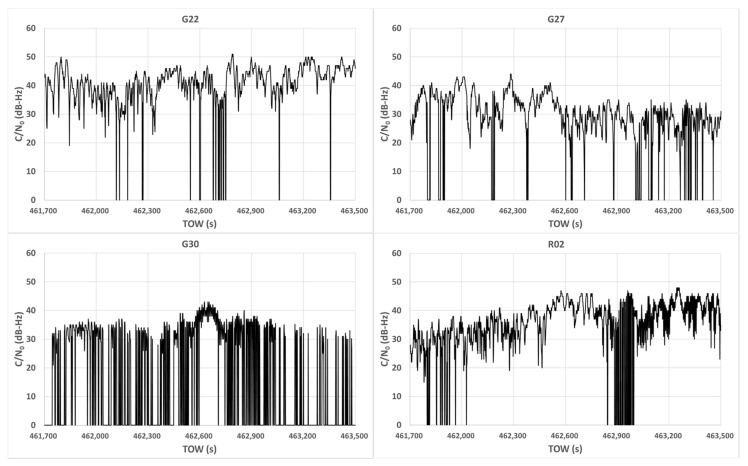
Temporal C/N_0_ mainly receiving non-line-of-sight (NLOS) signals.

**Figure 6 sensors-20-04059-f006:**
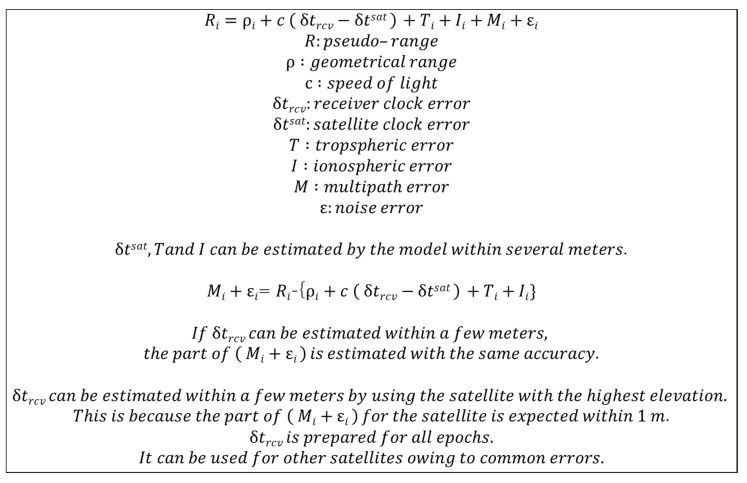
Summary and flow of pseudo-range error estimation.

**Figure 7 sensors-20-04059-f007:**
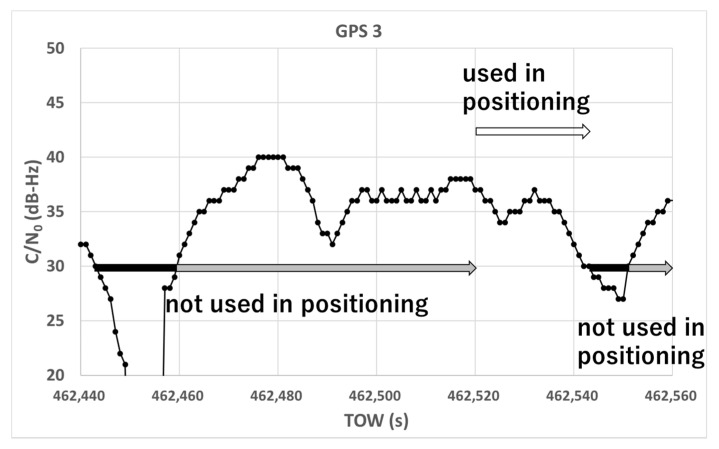
New concept for detecting multipath-contaminated satellites.

**Figure 8 sensors-20-04059-f008:**
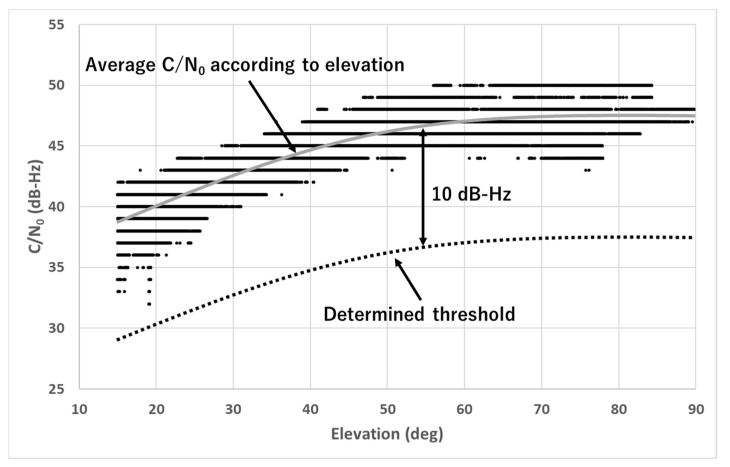
Relationship between elevation angle and C/N_0_, and proposed threshold (GPS L1-C/A).

**Figure 9 sensors-20-04059-f009:**
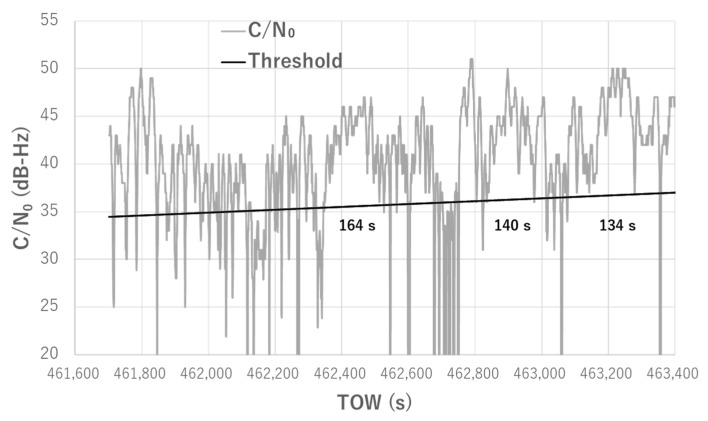
Real C/N_0_ for GPS 22 with elevation-dependent threshold and three longest periods.

**Figure 10 sensors-20-04059-f010:**
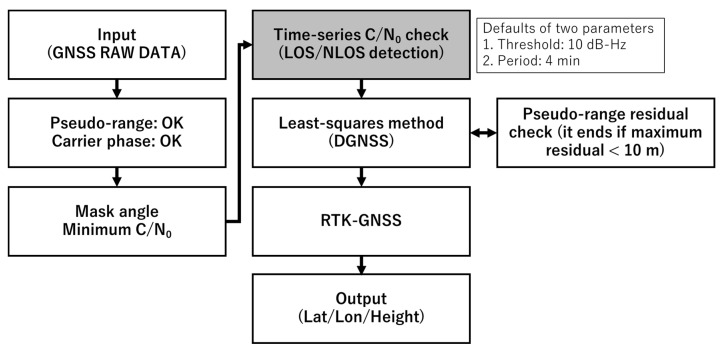
Flowchart of measurement check and positioning based on new concept.

**Figure 11 sensors-20-04059-f011:**
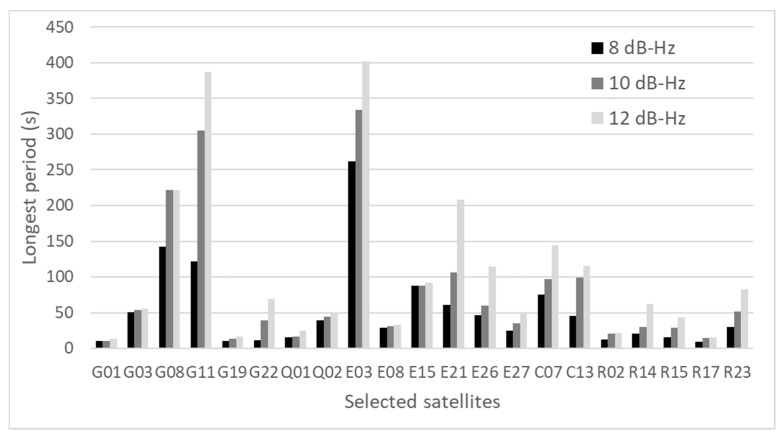
Longest period of selected satellites for three thresholds (high-rise buildings).

**Figure 12 sensors-20-04059-f012:**
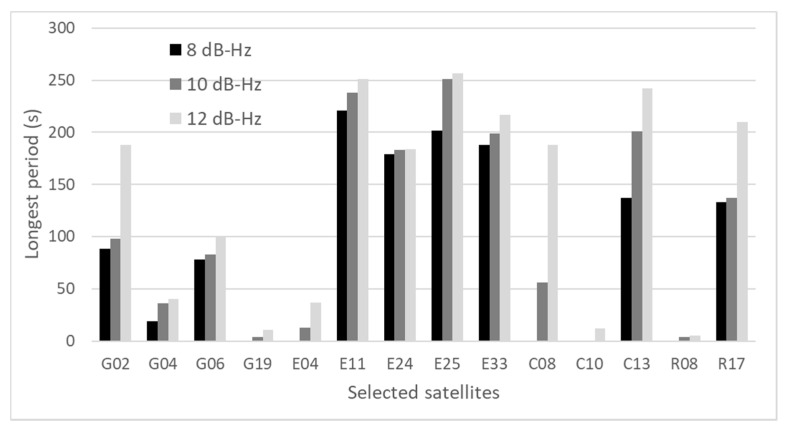
Longest period of selected satellites for three thresholds (medium-rise buildings).

**Figure 13 sensors-20-04059-f013:**
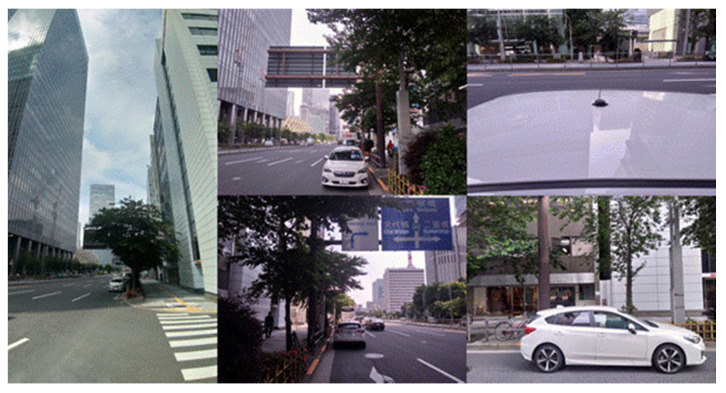
Detailed images around the antenna.

**Figure 14 sensors-20-04059-f014:**
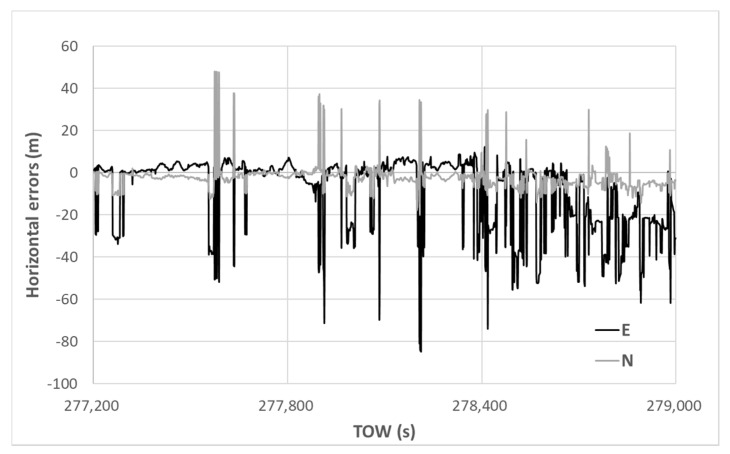
DGNSS horizontal errors.

**Figure 15 sensors-20-04059-f015:**
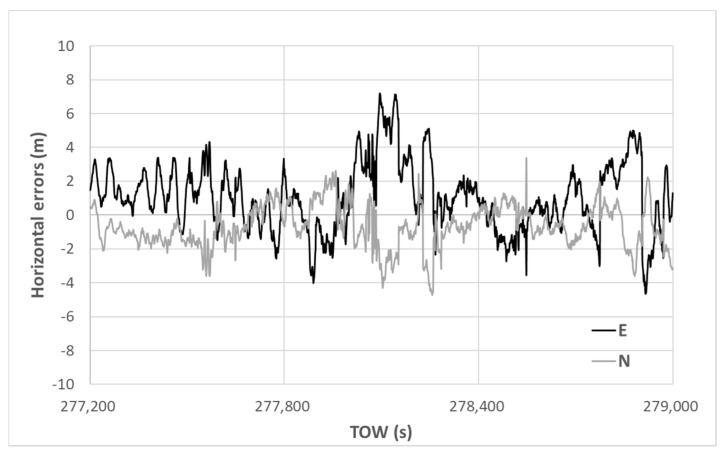
DGNSS horizontal errors using new method.

**Figure 16 sensors-20-04059-f016:**
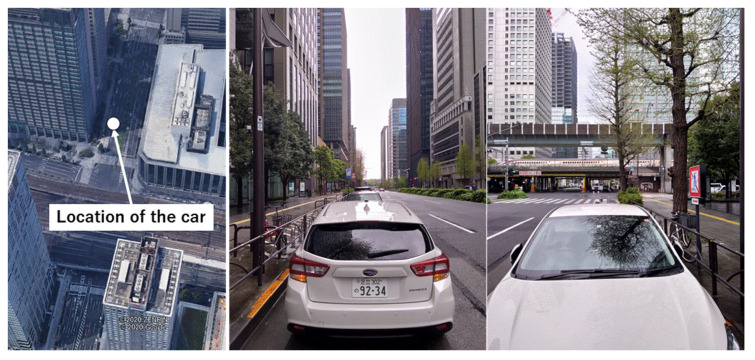
Location of car and detailed images around antenna.

**Figure 17 sensors-20-04059-f017:**
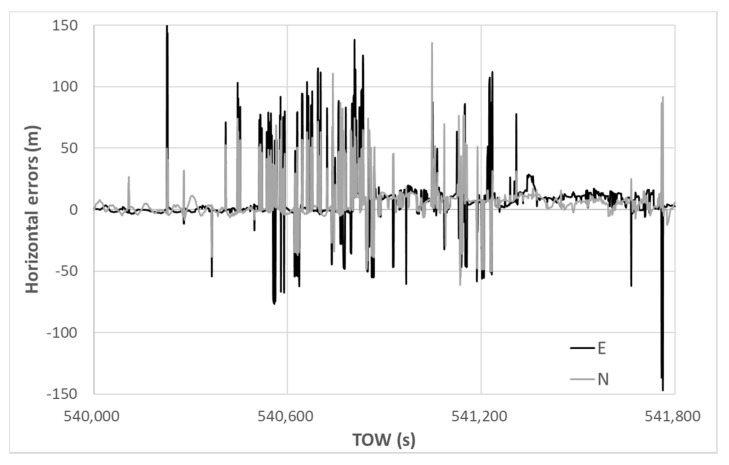
DGNSS horizontal errors.

**Figure 18 sensors-20-04059-f018:**
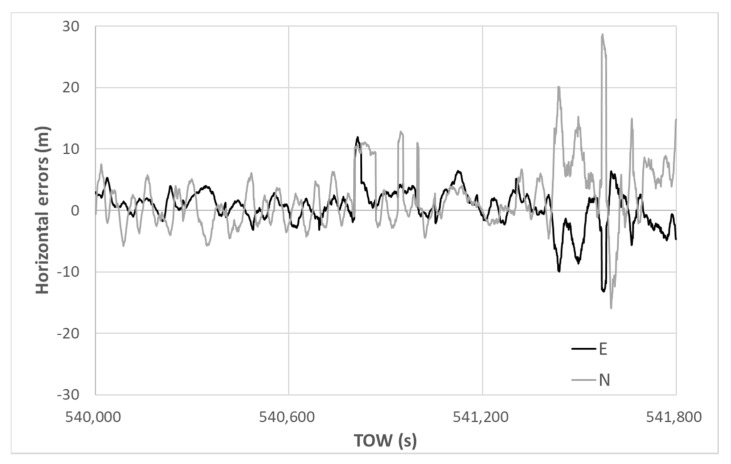
DGNSS horizontal errors using the new method.

**Figure 19 sensors-20-04059-f019:**
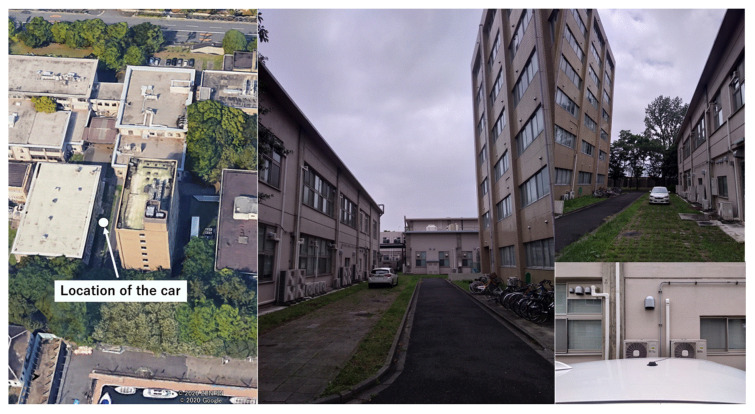
Location of car and detailed images around antenna.

**Figure 20 sensors-20-04059-f020:**
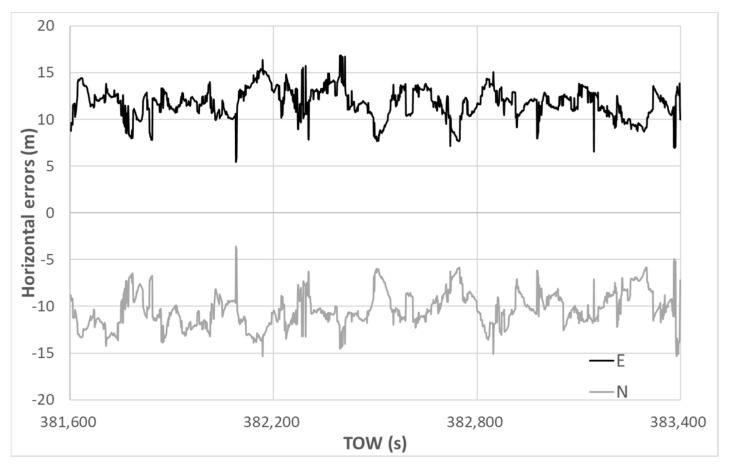
DGNSS horizontal errors.

**Figure 21 sensors-20-04059-f021:**
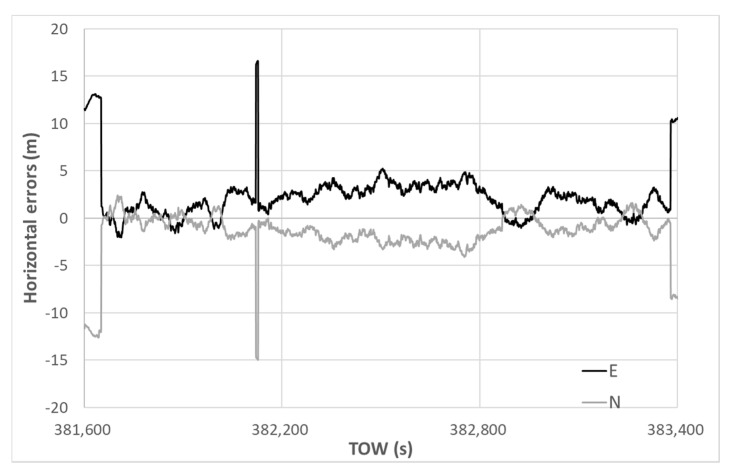
DGNSS horizontal errors using new method.

**Figure 22 sensors-20-04059-f022:**
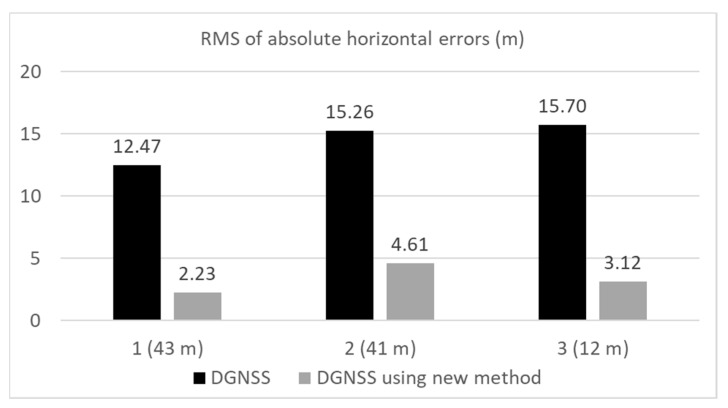
Comparison of DGNSS RMS errors in three locations.

**Figure 23 sensors-20-04059-f023:**
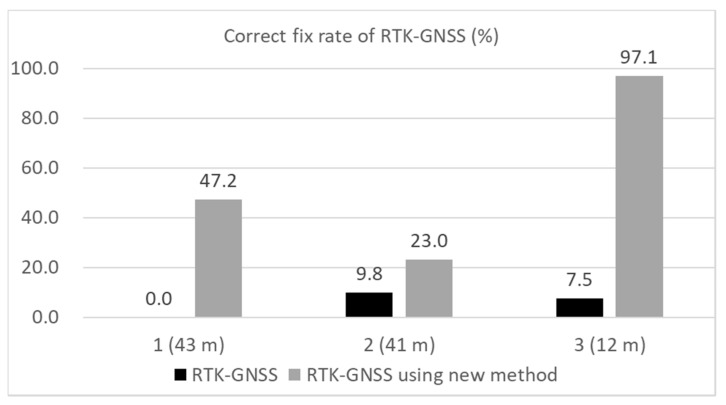
Comparison of RTK-GNSS correct fix rates in three locations.

**Figure 24 sensors-20-04059-f024:**
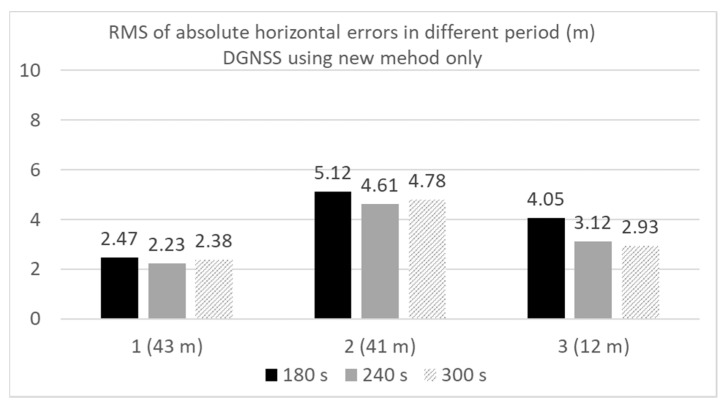
Comparison of DGNSS RMS errors in different periods.

**Figure 25 sensors-20-04059-f025:**
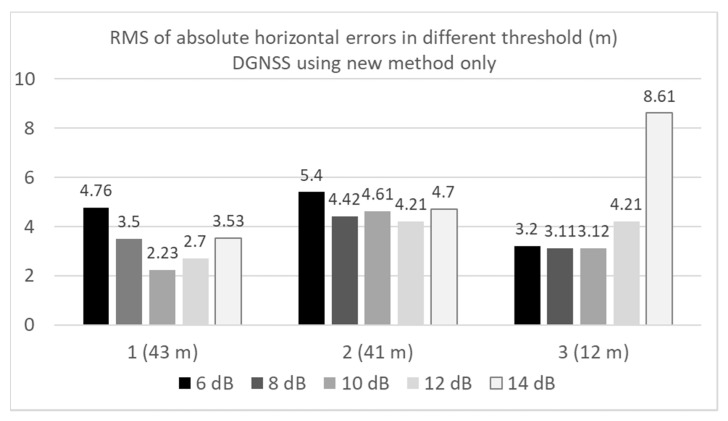
Comparison of DGNSS RMS errors in different thresholds.

**Table 1 sensors-20-04059-t001:** Statistics of pseudo-range errors and C/N_0._

SV	Flag	Maximum Error (m)	90th Percentile of Errors (m)	10th Percentile of Errors (m)	Percentage of C/N_0_ < 30 (%)	Percentage of C/N_0_ > 30 and C/N_0_ < 40 (%)	Percentage of C/N_0_ > 40 (%)
G01	LOS	1.5	1.1	0.6	0	0	100.0
G03	NLOS	233.9	150.5	65.0	16.4	74.2	4.1
G07	LOS	26.9	12.4	4.7	1.6	30.6	67.8
G08	NLOS	103.6	61.9	35.4	7.6	60.4	28.5
G11	LOS	5.2	4.1	1.1	1.0	35.4	63.6
G22	NLOS	155.3	99.6	56.5	2.3	31.2	64.1
G27	NLOS	556.4	111.4	77.4	19.9	38.3	6.7
G28	NLOS	339.6	73.8	11.4	37.7	19.7	0
G30	NLOS	320.8	90.5	35.5	4.5	39.3	3.3
J01	NLOS	279.0	175.5	116.6	31.0	53.3	0
J02	LOS	5.7	4.2	0.9	0	12.1	87.9
J03	LOS	0.1	0.1	0.1	0	0	100
J07	LOS	4.2	3.4	1.8	0	100	0
E01	LOS	7.9	5.6	0.7	0	4.1	95.9
E13	NLOS	198.1	121.1	39.0	68.3	30.1	0
E21	NLOS	210.7	115.2	47.5	13.3	79.0	6.9
E26	Partial	27.7	6.0	1.4	3.7	35.3	61.0
E31	NLOS	305.4	30.2	3.8	21.9	43.8	27.6
E33	NLOS	127.8	79.4	18.6	62.7	36.0	0
C07	LOS	3.8	2.7	0.7	0	0	100.0
C08	LOS	4.9	3.5	0.6	0	17.7	82.3
C10	LOS	11.8	4.1	0.5	4.2	38.5	43.7
C11	Partial	26.5	3.9	0.4	14.3	34.4	48.4
C13	LOS	13.5	11.3	5.5	2.4	31.6	66.0
C25	LOS	6.6	4.8	0.7	0	1.1	98.9
R02	NLOS	631.9	154.5	75.0	12.6	45.7	36.2
R15	LOS	8.2	6.4	3.2	0	0.6	99.4
R16	Partial	113.0	55.2	1.8	2.7	9.6	62.7
R17	LOS	9.8	7.3	4.3	0.2	3.7	96.1
R18	Partial	123.8	13.2	0.2	10.4	30.5	59.1
R19	NLOS	454.4	242.6	62.5	45.4	14.0	0

**Table 2 sensors-20-04059-t002:** Summary of continuous periods over threshold for selected satellites.

SV	90th Percentile of All Errors (m)	Percentage of C/N_0_ > 40 (%)	Longest Period (s)	Second Longest Period (s)	Third Longest Period (s)
G03	150.5	4.1	79	67	59
G08	61.9	28.5	89	64	63
G22	99.6	64.1	164	140	134
G27	111.4	6.7	149	127	86
G28	73.8	0	12	7	5
G30	90.5	3.3	108	13	13
J01	175.5	0	117	102	81
E13	121.1	0	85	63	54
E21	115.2	6.9	137	103	96
E31	30.2	27.6	17	-	-
E33	79.4	0	74	12	10
R02	154.5	36.2	163	130	109
R16	55.2	62.7	-	-	-
R19	242.6	0	42	40	31

**Table 3 sensors-20-04059-t003:** Configurations of sensors used in the test.

Sensor	Model Name
GNSS receiver	u-blox F9P (base/rover)
GNSS antenna (rover)	Standard patch antenna (ANN-MB-00-00)
GNSS antenna (base)	Trimble Zephyr 2 Geodetic

**Table 4 sensors-20-04059-t004:** Common parameters for satellite selection.

Item	Parameter
Mask angle	15 degrees
Maximum HDOP	10.0
Minimum C/N_0_ for L1 band	32 dB-Hz
Minimum C/N_0_ for L2 band	32 dB-Hz
Pseudo-range measurements	Tracked
Carrier phase measurements	Tracked
Carrier phase measurements (only RTK-GNSS)	Tracked and half-cycle resolved
Threshold for residual (least-squares method)	10.0 m

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
