# Peer review of "GNSS Multipath Detection Using Continuous Time-Series C/N0"

_sensors, 2020, doi:10.3390/s20144059_

Round 1
Reviewer 1 Report
Multipath is a time varing process in the urban environment for GNSS applicaions. This paper discussed this issue with static location test and get some initial results, which is not a new topic in this area but a tough topic as its difficulties for the movement signal processing.
In this paper, three cases static scenario were tested with different time averaging. In generally, the averaging could bring certain performance improvement as the random error might be eliminated partly. For the RTK application and even DGNSS, the user in mobile status and the long time averaging could not be done and then the paper conclusion will not be useful for this application.
Reviewer 2 Report
Review Summary and concerns:
This manuscript addresses a very interesting topic, the technique of multipath error detection by means of C/N_{0}. It is well written in all the sections, and the testing shows the impressive results in mitigating the multipath error. However, the data set used to give the identified conclusion in section 3, i.e. the relationship between between the pseudo-range error and C/N_{0}, is composed of 30-min observation at one testing point, this may be too few, or a lack of the powerful justification for the generalization of this data set. This work may become acceptable for publication after the concerns are addressed.
Concerns:
#45: INS, C/N_{0}, Please define the acronyms if firstly mentioned.
#79: QZS, GNS, A suspicious typo or new acronyms? What’s difference between GNSS satellite and GNS? #296: GLO, #301: GLONASS? Please keep the unique acronyms for them if they are the same. I suggest checking this case for the whole manuscript.
#113: The duration of the observation data is only 30 min of one day, it might be too few as the sample to test the presented technique. I suggest to test more data or give a strict justification for the generalization of the data.
#125-#128: “The four satellites GPS 01, QZSS 03, Galileo 01, and BDS 07 were selected”,“Moreover, the four satellites GPS 22, GPS 27, GPS 30, and GLONASS 02 were selected”, The strategy of the selection should be clarified.
#135: “The right-side buildings”, ”the tall building on the left side”, ..., the location description by means of “left/right” is confusing, the use of exact azimuth description regarding the target would be appreciated. I suggest checking this case for the whole manuscript.
#236-#244: Might they been consistent between the presented conclusion and the one by more case studies? The justification should be properly given.
#280-282. “This threshold ... 30 dB-Hz”, is there any precedent or more proof showing consistency with this “general” conclusion?
#285: “For example...”, The difference between GEO and other satellites is still not clear in this example.
#344: “32 dB-Hz”, why this minimum threshold of C/N_{0} is different with before, like 29, ~30? How do you get it? I don’t see the different conditions of them.
Reviewer 3 Report
Comments on sensors 857646
In this manuscript, the authors focused on the value of continuous time-series C/N0 for a certain period. Experimentally, they conducted three static tests at challenging locations near high-rise buildings in Tokyo. The proposed method substantially mitigates multipath errors in the differential GNSS by removing the NLOS signals appropriately. Consequently, the performance of real-time kinematic GNSS is improved significantly. The manuscript is well written and concludes some novel results, and the theory and experiment are well performed in a satisfactory way. Thus, I recommend this paper publication in sensors. However, despite the nature of the interesting results reported in this study, there are several issues in the paper need to be elucidated further.
- To clear demonstrate the information, some figures(Fig.4, Fig.13 and Fig.16) should redraw to magnify the vertical coordinates to let the readers see the fluctuations and the details.
- The conclusions are a summary of previously stated in abstract and introduction and do not include concluding arguments.
- The authors should mention some related references to demonstrate a whole and completed background and discussion in the part of introduction. For instance, the reference [APL, 2020, 116 (1):011101] should be added.
Reviewer 4 Report
The paper has potential and addresses a still-relevant problem in GNSS but in its current form it lacks focus and clarity. Several major and minor issues need to be fixed before publication.
The authors state that Fig 4 shows satellites mainly in LOS and Fig 5 shows satellites mainly in NLOS: how did the authors determine whether a satellite is in LOS or NLOS? Based on the elevation angle? Based on something else?
In Line 173, the authors state: “According to the flow presented in Figure 6, we estimated the receiver clock error by using a reliable QZS.” What does a ‘reliable QZS’ mean? How was the reliability validated? How many satellites from QZS were used? Why is the QZS more reliable than GPS?
The purpose of the paper stated in the abstract does not match the experimental work from Section 2.2. Was the purpose to analyze GNSS /GPS or differential GNSS (DGNSS). Abstract talks about GNSS; experiments talk about DGNSS. This is confusing and must be clarified.
Clarifications should be added in Section 2.3 regarding what is considered NLOS and what is considered LOS and why. Do all satellites with elevation above 15 degrees are considered as ‘LOS’ satellites? What about the effects of the tall buildings, how are they taken into account? Without a realistic benchmark of what consists as NLOS and what consists as LOS scenarios, the results lack meaning. A clear table and/or rule must be added to explain how one knows if a satellite is LOS or NLOS. Also, how the authors take the dynamics into account? Measurements are collected over 30’, interval over which one satellite which was initially LOS can become NLOS or vice versa. Can a flag be added in Table 1 to show the conditions LOS/NLOS/partial LOS of each satellite listed in column 1?
What does it mean “used” and “not used” in Fig 7? What parameter is used/not used and for what purpose? Explanations are missing in the text.
Fig. 8 lacks a legend to explain the different curves in it
Table 2 is unclear: what are the errors in meters per satellite? They are not corresponding to the receiver error as this cannot be computed per satellite. If they correspond to NLOS error, how was the ground truth computed?
A block diagram or a pseudocode of the proposed algorithm should be added to explain the meaning of ‘longest period’, ‘second longest period’, etc and why they are relevant to be used as metrics (eg see Table 2 metrics) – eg improving the flowchart in Fig 10 by adding the steps where the periods (longest, second longest, etc) are computed. The flow chart in Fig 10 should also clarify what is the output, as now it ends with a residual check and it misses the final information – how one decides if we are in LOS or NLOS situation?
The threshold choice for C/N0 seems to be fully empirical and relying on a off-line calibration phase with measurements at the desired location. A discussion should be added about whether the thresholds can be selected automatically, without any offline measurements and how much the threshold choice influences the results. For example, could it be that with an unsuitable threshold one gets even higher errors than with traditional RTK-GNSS?
Round 2
Reviewer 1 Report
This revised version provides more analysis on the test data and the potential applications of this methodology, It seems that the paper might be published with this style.
Reviewer 4 Report
The authors addressed in satisfactory manner all review concerns